# Learned Thresholds Token Merging and Pruning for Vision Transformers

**Maxim Bonnaerens**                                    *maxim.bonnaerens@ugent.be*
*IDLab-AIRO, Ghent University - imec*

**Joni Dambre**                                         *joni.dambre@ugent.be*
*IDLab-AIRO, Ghent University - imec*

**Reviewed on OpenReview:** `https://openreview.net/forum?id=WYKTCKpImz`

## Abstract

Vision transformers have demonstrated remarkable success in a wide range of computer vision tasks over the last years. However, their high computational costs remain a significant barrier to their practical deployment. In particular, the complexity of transformer models is quadratic with respect to the number of input tokens. Therefore techniques that reduce the number of input tokens that need to be processed have been proposed. This paper introduces Learned Thresholds token Merging and Pruning (LTMP), a novel approach that leverages the strengths of both token merging and token pruning. LTMP uses learned threshold masking modules that dynamically determine which tokens to merge and which to prune. We demonstrate our approach with extensive experiments on vision transformers on the ImageNet classification task. Our results demonstrate that LTMP achieves state-of-the-art accuracy across reduction rates while requiring only a single fine-tuning epoch, which is an order of magnitude faster than previous methods. Code is available at `https://github.com/Mxbonn/ltmp`.

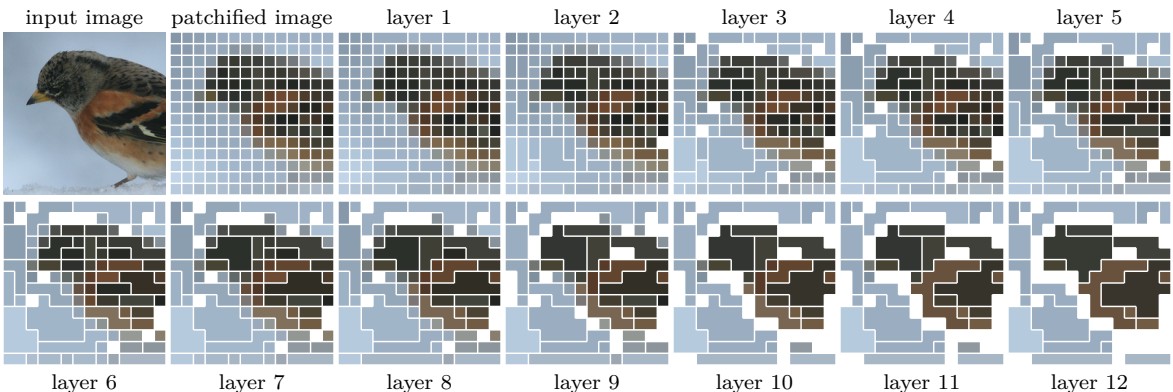

Figure 1: Visualization of the merging and pruning as applied to image patches. In every layer the most similar tokens are merged and any unimportant tokens are pruned. The visualizations show the remaining tokens after every layer in DeiT-S.

## 1    Introduction

The adoption of transformers (Vaswani et al., 2017), originally developed for natural language processing, in the field of computer vision with Vision Transformers (ViT) (Dosovitskiy et al., 2021) has led to significant progress in the field. But despite the impressive results of vision transformers, their success has come with

a cost; ViT models are computationally expensive and require larger datasets (e.g. ImageNet-21k instead of ImageNet-1k (Deng et al., 2009)) and prolonged training times (Dosovitskiy et al., 2021). In order to benefit from their accuracy in downstream tasks and applications, the use of pretrained models has become essential. However, their adoption on resource-constrained devices such as mobile and embedded platforms remains limited due to the high computational cost.

To reduce the computational cost of transformers, previous work has focused on techniques such as distillation (Wu et al., 2022), quantization (Liu et al., 2021b) and pruning. Pruning techniques have explored pruning model weights (Gordon et al., 2020), attention heads (Voita et al., 2019), and input tokens (Rao et al., 2021). This last approach is very effective, as the model complexity of a transformer is quadratic to the number of tokens. In vision transformers, the tokens are non-overlapping patches of an image, e.g. a token may represent patches of $16 \times 16$ pixels. Token pruning has attracted research interest as it matches our intuition that not all parts of an image are equally important. The self-attention mechanism in transformers, due to its ability to process variable length inputs and its order-agnostic characteristic, enables unstructured reduction of the number of tokens between layers. This was previously non-trivial with convolutional networks.

Most token pruning approaches calculate an importance score for every token in each layer and remove the least important tokens. While token pruning has been shown to be an effective compression technique, removing tokens results in loss of information which limits the amount of tokens that can be pruned. In order to recover from this loss of information, most pruning approaches require substantial retraining to be effective. Additionally, some recent pruning techniques have incorporated token combining techniques where the pruned tokens are combined into a single token that aggregates the information that would otherwise be lost (Kong et al., 2022).

Token merging (ToMe) (Bolya et al., 2023) takes this combining technique one step further. ToMe exclusively combines pairs of tokens into new tokens rather than pruning them. This has as advantage that it does not discard but summarizes information, leading to better accuracy while being equally effective in reducing computational complexity.

In this work, we introduce Learned Thresholds token Merging and Pruning (LTMP). Our approach combines the benefits of token merging, which allows us to combine rather than discard token information, with pruning, which allows us to remove uninformative tokens. To the best of our knowledge, this is the first work to extensively combine these two reduction techniques, which leads to improved accuracy compared to previous work. Our approach uses learned threshold masking modules, which allow the model to learn thresholds that determine which tokens to prune and which ones to merge in each layer. This enables adaptive token reduction, while only requiring two learnable parameters per transformer block. As a result, our approach converges within a single epoch, reducing the fine-tuning cost by an order of magnitude compared to other learnable pruning approaches.

Our contributions can be summarized as follows:

- We propose to combine token merging with token pruning, enabling us to achieve high token reduction rates with minimal loss of accuracy.

- Our method introduces learned threshold masking modules, which require only two learnable parameters per transformer block, allowing our approach to converge within a single epoch.

- We optimize the thresholds using a novel budget-aware training loss for which we introduce a reduction target $r_{target}$ and an actual FLOPs reduction factor $r_{FLOPs}$. This allows us to create models of any size and allows the model to freely distribute the reduction operations across layers.

## 2 Related work

### 2.1 Efficient Vision Transformers

Initially, transformers (Vaswani et al., 2017) were adopted from NLP to computer vision (Dosovitskiy et al., 2021) for their impressive accuracy. But despite their success in many vision tasks, ViT-based models could

not compete with lightweight CNNs for deployment on resource-constrained platforms (Wang et al., 2022). To create efficient ViT models, several architecture changes have been proposed which modify the expensive attention mechanisms (Kitaev et al., 2019; Chen et al., 2021; Liu et al., 2021a; Li et al., 2021). In this paper, we look at token pruning, but other pruning approaches that have been successfully applied to transformers are, among others, weight pruning (Gordon et al., 2020) and attention heads pruning (Voita et al., 2019).

### 2.2 Token Pruning

The flexibility of transformers with respect to the sequence length and order of the inputs allows token pruning, something which was previously non-trivial to do in convolutional-based models. Token pruning methods can differ in various ways, such as the score used to determine the importance of each token. Pruning methods can also differ in the way that token reduction is applied. In fixed rate pruning (Goyal et al., 2020; Rao et al., 2021; Bolya et al., 2023; Liang et al., 2022; Xu et al., 2022) a predefined number of tokens is removed per layer, while in adaptive approaches (Kim et al., 2022; Yin et al., 2021; Liu et al., 2022) the tokens are pruned dynamically based on the input.

The most recent pruning approaches do not only prune tokens but they also create a single additional token after each pruning step. This token aggregates the information of the pruned tokens and limits the loss of accuracy while pruning. EViT (Liang et al., 2022), Evo-ViT (Xu et al., 2022) and SPViT (Kong et al., 2022) use a weighted average based on the importance score to create the new fused token.

### 2.3 Token merging

Token Merging (ToMe), as introduced by Bolya et al. (2023), introduces a lightweight token matching algorithm to merge similar tokens. It is as fast as pruning while being more accurate. ToMe partitions all tokens into two sets $\mathbb{A}$ and $\mathbb{B}$ of roughly equal size by alternating and calculates similarity scores for each token in $\mathbb{A}$ with every token in $\mathbb{B}$. The similarity score is defined as the cosine similarity between the key vectors ($\mathbf{K}$) used in the self-attention layer. The final similarity score of a token in $\mathbb{A}$ is the highest similarity score with any token in $\mathbb{B}$. Based on this score, ToMe merges the $k$ most similar tokens through averaging and concatenates the two sets back together.

## 3 Learned thresholds token merging and pruning

### 3.1 Overview

An overview of our framework is shown in Figure 2. Given any vision transformer, our approach adds merging (LTM) and pruning (LTP) components with learned threshold masking modules in each transformer block between the Multi-head Self-Attention (MSA) and MLP components. Based on the attention in the MSA, importance scores for each token and similarity scores between tokens are computed. Learned threshold masking modules then learn the thresholds that decide which tokens to prune and which ones to merge.

### 3.2 Motivation

Although token merging is generally more accurate than pruning as it combines tokens instead of discarding them, it is not always better to merge tokens instead of discarding them. In some cases, it may be more beneficial to prune an unimportant token rather than merging the most similar tokens, as the similarity between them may not be very high.

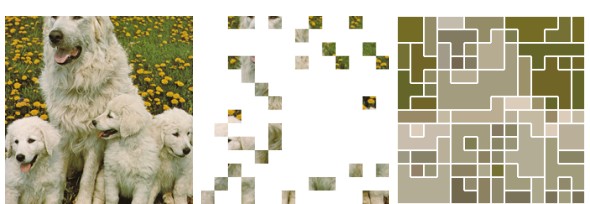

Figure 3: Visualization of token pruning (middle) compared to token merging (right). The visualization shows the remaining tokens after the 10-th layer in ViT-B when pruning or merging 16 tokens per layer.

In this section, we explore whether token merging and token pruning are techniques that can be combined. Figure 3 visualizes tokens kept by pruning and by merging on one specific image, we observe that the kept tokens are noticeably different between

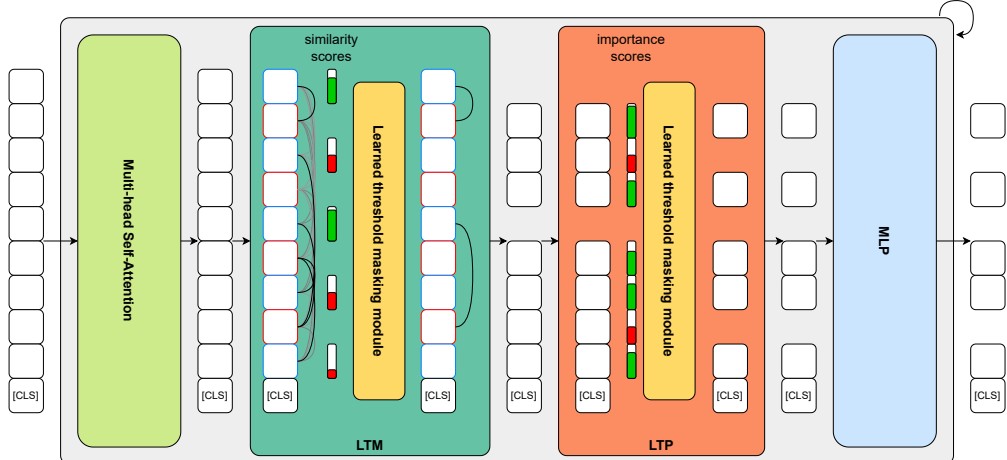

Figure 2: Overview of our approach. LTMP contains a merging and pruning component, each with a learned threshold masking module. The components are added between the Multi-head Self-Attention and MLP components of each transformer block.

both approaches. To quantify the relation between merging and pruning, we calculated the Kendall tau rank correlation (Kendall, 1938) between the *importance scores* used in token pruning and the *similarity scores* used in token merging. However, as merging only calculates similarity scores for tokens in set $\mathbb{A}$ (see Section 2.3), we compute the Kendall tau correlation between the similarity scores and the importance scores of only the tokens in set $\mathbb{A}$. We calculated the correlations over 1000 images using a ViT-B where each layer pruned and merged 8 tokens in a fixed top-$k$ manner and report the results in Table 1. We find that the $\tau$ correlations between both scores are low, especially in the early layers where the most important merging and pruning is done. We, therefore, propose combining token merging and token pruning.

Table 1: Kendall $\tau$ correlation between *importance scores* in token pruning and *similarity scores* in token merging. The correlations are calculated over 1000 images using a ViT-B where each layer pruned and merged 8 tokens.

| layer | 1 | 3 | 5 | 7 | 9 | 11 |
|---|---|---|---|---|---|---|
| $\tau$ | 0.01 | 0.14 | 0.21 | 0.19 | 0.15 | 0.22 |

### 3.3 Learned thresholds

#### 3.3.1 Learned thresholds pruning

Our learned thresholds approach is conceptually similar to learned token pruning as introduced in Kim et al. (2022). In each transformer block an *importance score* is calculated for every token $\mathbf{x}_i, i \in \{1, ..., n\}$, where $n = hw$ is the number of tokens[1]. A threshold $\theta^l \in \mathbb{R}, l \in \{1, ..., L\}$, where $L$ is the number of transformer blocks, determines which tokens to keep and which to prune in each layer; only tokens with an importance score above the threshold are kept.

In order to prune tokens adaptively, we introduce a threshold masking module that, given the importance scores $\mathbf{s}^l \in \mathbb{R}^n$, learns a pruning threshold $\theta^l$ and outputs which tokens to keep.

$$M(\mathbf{s}^l_i, \theta^l) = \begin{cases} 1, & \text{if } \mathbf{s}^l_i > \theta^l \\ 0, & \text{otherwise} \end{cases} \tag{1}$$

---

[1]We omit the [CLS] class token for simplicity, during pruning and/or merging we always keep the [CLS] token.

However, in order to make $\theta^l$ learnable during training, the threshold masking module needs to be differentiable. We achieve this by implementing the threshold masking module as a straight-through estimator (Bengio et al., 2013), where we estimate the masking function during backpropagation as

$$M(\mathbf{s}_i^l, \theta^l) = \sigma\left(\frac{\mathbf{s}_i^l - \theta^l}{\tau}\right) \tag{2}$$

where $\sigma(x)$ is the sigmoid function and $\tau$ is the temperature hyperparameter.

During inference we only keep the tokens in the $l$-th block where $M(\mathbf{s}_i^l, \theta^l) = 1$. However, during training, we can not simply drop tokens as that does not allow the model to backpropagate the influence of the threshold on the model performance. We, therefore, create a mask indicating which tokens are kept and which ones are pruned. Every threshold masking module only updates the entries of the mask for the tokens that have not yet been removed prior to that layer, as tokens that are pruned in an earlier layer have to remain pruned. We construct the pruning mask $\mathbf{m}^l \in [0,1]^n$ as follows:

$$\mathbf{m}_i^l = \begin{cases} M(\mathbf{s}_i^l, \theta^l), & \text{if } \mathbf{m}_i^{l-1} = 1 \\ \mathbf{m}^{l-1}, & \text{otherwise} \end{cases} \tag{3}$$

The learned token pruning implementation in Kim et al. (2022) multiplies its mask with the tokens in order to create zero valued tokens. However, these tokens do not remain zero due to bias terms in MLP layers; furthermore adding zero valued tokens changes attention calculations compared to removing those tokens. Instead, our approach makes changes to the only place where tokens influence each other: the attention mechanism. [2]

Recall the original formula for attention (Vaswani et al., 2017):

$$\text{Attention}(\mathbf{Q}, \mathbf{K}, \mathbf{V}) = \text{softmax}\left(\frac{\mathbf{Q}\mathbf{K}^T}{\sqrt{d_k}}\right)\mathbf{V} \tag{4}$$

In order to avoid that the masked tokens influence the attention mechanism, we propose a modified function:

$$\text{Attention\_with\_mask}(\mathbf{Q}, \mathbf{K}, \mathbf{V}, \mathbf{m}) = \mathbf{S}\mathbf{V} \tag{5}$$

where,

$$\mathbf{S}_{ij} = \frac{\exp(\mathbf{A}_{ij})\mathbf{m}_j}{\sum_{k=1}^{N} \exp(\mathbf{A}_{ik})\mathbf{m}_k}, 1 \leq i, j, k \leq n \tag{6}$$

and,

$$\mathbf{A} = \mathbf{Q}\mathbf{K}^T / \sqrt{d_k} \in \mathbb{R}^{n \times n} \tag{7}$$

Equation (6) computes a masked softmax, which is equivalent to a softmax calculated with the pruned tokens removed. Attention\_with\_mask is conceptually similar to the masked attention as found in the transformer decoder of language models. However, where the masking in transformer decoders is done by setting masked tokens to $-\infty$, our approach requires the influence of the straight-through estimator mask to propagate to the thresholds during backpropagation.

### 3.3.2 Learned thresholds merging

Token merging is originally a top-$k$ approach, meaning that it merges based on a fixed rate and has no learnable parameters. We modify ToMe to use thresholds instead of top-$k$ by applying the same techniques as introduced in Section 3.3.1; this is by adding our learned threshold masking module, in which similarity scores above these thresholds are selected for merging, and by changing the attention function to Equation (5).

---

[2]Technically, tokens also influence each other during layer normalization, however as pruning is done on pretrained models, we simply use the global statistics from pretraining during normalization.

### 3.3.3 Learned thresholds merging and pruning

With learnable thresholds, it is trivial to combine merging and pruning, as we can simply add a learned threshold masking module that learns thresholds for importance scores and another module that learns thresholds for similarity scores.

## 3.4 Training Strategy

### 3.4.1 Training objective

To effectively reduce the number of tokens in the transformer blocks, it is necessary to include a regularization loss term in the training process. Without this loss, the model has no incentive to prune any tokens and the pruning thresholds will simply be set to 0 as the most accurate model uses all inputs. We propose a budget-aware training loss which introduces a reduction target $r_{\text{target}}$ for the FLOPs of the vision transformer.

Let us denote $\phi_{\text{module}}(n, d)$ as a function that calculates the FLOPs of a module based on the number of tokens $n$ and the embedding dimension $d$. The actual FLOPs reduction factor $r_{\text{FLOPs}}$ of a ViT can then be computed as:

$$r_{\text{FLOPs}} = \frac{\phi_{\text{PE}}(n,d)}{\phi_{\text{ViT}}(n,d)} + \sum_{l=1}^{L} \frac{\phi_{\text{BLK}}(n,d)}{\phi_{\text{ViT}}(n,d)} \left( \frac{\phi_{\text{MSA}}(\bar{\mathbf{m}}^{l-1}n, d)}{\phi_{\text{BLK}}(n,d)} + \frac{\phi_{\text{MLP}}(\bar{\mathbf{m}}^{l}n, d)}{\phi_{\text{BLK}}(n,d)} \right) + \frac{\phi_{\text{HEAD}}(\bar{\mathbf{m}}^{l}n, d)}{\phi_{\text{ViT}}(n,d)} \tag{8}$$

where $\bar{\mathbf{m}}^{l} = \frac{1}{n}\sum_{i=1}^{n}\mathbf{m}_i^l$ is the percentage of input tokens that are kept after the $l$-th threshold masking operation and $\bar{\mathbf{m}}^0 = 1$. PE, BLK and HEAD denote the different components of a vision transformer: the patch embedding module, the transformer blocks and the classification head.

As the vast majority of the FLOPs in a vision transformer occurs in the transformer blocks ($\approx 99\%$ in ViT-S), we ignore the FLOPs in the patch embedding and classification head: $\frac{\phi_{\text{PE}}(n,d)}{\phi_{\text{ViT}}(n,d)} = \frac{\phi_{\text{HEAD}}(n,d)}{\phi_{\text{ViT}}(n,d)} \approx 0$. That means that we can simplify

$$\frac{\phi_{\text{BLK}}(n,d)}{\phi_{\text{ViT}}(n,d)} \approx \frac{1}{L}, \tag{9}$$

where $L$ is the number of transformer blocks.

The FLOPs of a transformer block and its two components, the MSA and MLP can be computed as:

$$\phi_{\text{MSA}}(n, d) = 4nd^2 + 2n^2d \tag{10}$$

$$\phi_{\text{MLP}}(n, d) = 8nd^2 \tag{11}$$

$$\phi_{\text{BLK}}(n, d) = \phi_{\text{MSA}}(n, d) + \phi_{\text{MLP}}(n, d) = 12nd^2 + 2n^2d \tag{12}$$

Substituting Equations (10) to (12) into Equation (8) gives:

$$r_{\text{FLOPs}} \approx \sum_{l=1}^{L} \frac{1}{L} \left( \frac{2\bar{\mathbf{m}}^{l-1}nd^2 + (\bar{\mathbf{m}}^{l-1}n)^2 d + 4\bar{\mathbf{m}}^l nd^2}{6nd^2 + n^2d} \right) \tag{13}$$

Given this FLOPs reduction factor $r_{\text{FLOPs}}$ as a function of the threshold masks, we define our regularization loss as the squared error between the reduction target and the actual FLOPs reduction factor:

$$\mathcal{L}_{\text{reg}} = (r_{\text{target}} - r_{\text{FLOPs}})^2 \tag{14}$$

This regularization loss is then combined with the classification loss, for which we adopt the standard cross entropy loss.

$$\mathcal{L} = \mathcal{L}_{\text{CE}} + \lambda\mathcal{L}_{\text{reg}} \tag{15}$$

The overall training objective is to learn thresholds that optimize the model while reducing the model complexity to a certain reduction target. The combination of learned thresholds and our budget-aware loss enables the model to optimally distribute merging and pruning across layers.

### 3.4.2 Training schedule

LTMP only adds two learnable parameters per transformer block (one for pruning and one for merging). As is common in pruning it is applied to pretrained models. We therefore only update the thresholds during training and keep all other trainable parameters fixed, allowing LTMP to converge within a single epoch.

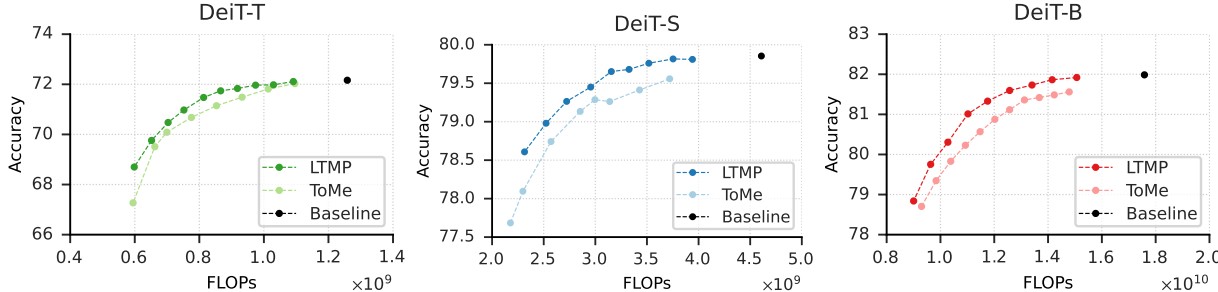

Figure 4: Accuracy/FLOPs trade-offs for LTMP variants of DeiT-Tiny, -Small and -Base. FLOPs are plotted in logarithmic scale. ToMe accuracy/FLOPs trade-offs are shown as a comparison.

## 4 Experiments

In this section, we demonstrate our approach through extensive experiments on the ImageNet-1k classification task (Deng et al., 2009) using various ViT variants (Steiner et al., 2022; Touvron et al., 2021). All pretrained models are taken from the `timm` PyTorch library (Wightman, 2022).

All our experiments are trained for a single epoch, using SGD without momentum and a batch size of 128. The remaining training settings such as augmentations are set to the default values of `timm`. The hyperparameters that are introduced in LTMP are set to $\tau = 0.1$ and $\lambda = 10$. As the importance scores and similarity scores have values in different ranges we use separate learning rates for the thresholds in the pruning modules and the merging modules: $5 \cdot 10^{-6}$ for the pruning thresholds and $5 \cdot 10^{-3}$ for the merging thresholds. To select these hyperparameters, we used a separate held-out validation set containing 2% of the original ImageNet training set. The final hyperparameters are determined based on their performance on DeiT-Small with $r_{target}$ set to 0.65. To show the robustness of these hyperparameters, we did not change them between model variations and reductions targets (i.e. ViT-Tiny with $r_{target}$ set to 0.45, is fine-tuned with the same hyperparameters).

### 4.1 Main results

As most other pruning approaches require extensive fine-tuning, the most commonly used baseline vision transformer in other works is the data-efficient vision transformer DeiT (Touvron et al., 2021), for this reason, we also report on DeiT models.

In Figure 4, we show our approach applied to DeiT-Tiny, -Small and -Base. For each model we vary $r_{target}$ such that we obtain a set of reduced models, each with a different model size. Table 2 lists the detailed results for the DeiT models. Models used in one of the other tables in the paper are highlighted in gray . We included $r_{\text{target}}$, such that our results can be reproduced. In Appendix A, we additionally included results on the standard ViT models. All accuracy numbers are reported with 3 significant figures; a brief analysis of the error ranges on the accuracy numbers can be found in Appendix B.

### 4.2 Comparison to other work

In Table 3, we compare the reported top-1 accuracy, FLOPs[3], and fine-tune epochs of other pruning approaches to our work. The other approaches we compare with are SPViT (Kong et al., 2022), DynamicVit

---

[3]We follow the convention of reporting FLOPs as multiply-adds.

Table 2: Detailed results for DeiT models.

| Model | $r_{\text{target}}$ | FLOPs | Accuracy |
|---|---|---|---|
| DeiT-T (Baseline) | | 1.258G | 72.2 |
| DeiT-T | 0.85 | 1.091G | 72.1 |
| DeiT-T | 0.80 | 1.029G | 72.0 |
| DeiT-T | 0.75 | 0.974G | 72.0 |
| DeiT-T | 0.70 | 0.918G | 71.8 |
| DeiT-T | 0.65 | 0.866G | 71.7 |
| DeiT-T | 0.60 | 0.813G | 71.5 |
| DeiT-T | 0.55 | 0.752G | 71.0 |
| DeiT-T | 0.50 | 0.703G | 70.5 |
| DeiT-T | 0.45 | 0.652G | 69.8 |

| Model | $r_{\text{target}}$ | FLOPs | Accuracy |
|---|---|---|---|
| DeiT-S (Baseline) | | 4.608G | 79.9 |
| DeiT-S | 0.85 | 3.940G | 79.8 |
| DeiT-S | 0.80 | 3.753G | 79.8 |
| DeiT-S | 0.75 | 3.518G | 79.8 |
| DeiT-S | 0.70 | 3.326G | 79.7 |
| DeiT-S | 0.65 | 3.155G | 79.7 |
| DeiT-S | 0.62 | 3.016G | 79.6 |
| DeiT-S | 0.60 | 2.955G | 79.5 |
| DeiT-S | 0.55 | 2.722G | 79.3 |
| DeiT-S | 0.50 | 2.523G | 79.0 |
| DeiT-S | 0.45 | 2.314G | 78.6 |

| Model | $r_{\text{target}}$ | FLOPs | Accuracy |
|---|---|---|---|
| DeiT-B (Baseline) | | 17.583G | 81.99 |
| DeiT-B | 0.85 | 15.007G | 81.9 |
| DeiT-B | 0.80 | 14.152G | 81.9 |
| DeiT-B | 0.75 | 13.403G | 81.7 |
| DeiT-B | 0.70 | 12.571G | 81.6 |
| DeiT-B | 0.65 | 11.754G | 81.3 |
| DeiT-B | 0.60 | 11.022G | 81.0 |
| DeiT-B | 0.55 | 10.276G | 80.3 |
| DeiT-B | 0.50 | 9.608G | 79.4 |
| DeiT-B | 0.45 | 9.003G | 78.8 |

(Rao et al., 2021), EViT (Liang et al., 2022), EvoViT (Xu et al., 2022) and ToMe (Bolya et al., 2023). Most works report on a pruned model with around 3.0 GFLOPs, which for DeiT-S corresponds to $r_{target} \approx 0.65$ in our approach. Only SPViT reports on a different model size, which is why it is compared separately. The results show that LTMP reaches state-of-the-art accuracy at a fraction of the fine-tuning epochs required by other learnable methods. The accuracy of LTMP matches or exceeds the accuracy of other token pruning approaches which require a minimum of 30 fine-tune epochs. Only EViT is able to reach a higher accuracy than LTMP, but only when drastically increasing the fine-tune epochs to 100, which is two orders of magnitude more than our approach. Because ToMe requires no fine-tuned checkpoints for comparisons, we are able to compare LTMP to ToMe more extensively over a wide range of model sizes. Figure 4 shows the accuracy/FLOPs trade-offs for LTMP and ToMe. Our experiments show that LTMP consistently outperforms ToMe across model sizes.

Table 3: Comparison to other token reduction approaches on DeiT-S. Our method reaches state-of-the-art accuracy with significantly fewer fine-tune epochs than other learnable approaches.

| Method | FLOPs | Accuracy | fine-tune epochs |
|---|---|---|---|
| DeiT-S (Baseline) | 4.6G | 79.8 | - |
| SPViT | 3.8G | 79.8 | 75 |
| **LTMP (Ours)** | 3.8G | 79.8 | 1 |
| DynamicViT | 2.9G | 79.3 | 30 |
| EViT | 3.0G | 79.5 | 30 |
| EViT | 3.0G | 79.8 | 100 |
| Evo-ViT | 3.0G | 79.4 | 300 |
| ToMe | 3.0G | 79.3 | 0 |
| **LTMP (Ours)** | 3.0G | 79.6 | 1 |

## 4.3 Synergy between merging and pruning

In order to analyze the synergy between token merging and token pruning, we have examined the distribution of merged and pruned tokens across each layer of the vision transformer. The results, shown in Figure 5 for DeiT-S, reveal that token merging is the dominant reduction operation in the early layers of the transformer, while token pruning is more prevalent in the final layers. This aligns with the findings of Bolya et al. (2023) that merging is more effective than pruning, as it can summarize information. However, once all similar tokens are merged, it is more beneficial to prune the least informative tokens rather than merging tokens that are not as similar. In other words, the first layers are mainly used to combine similar patches and once this is mostly done, token pruning removes the unimportant parts of the input.

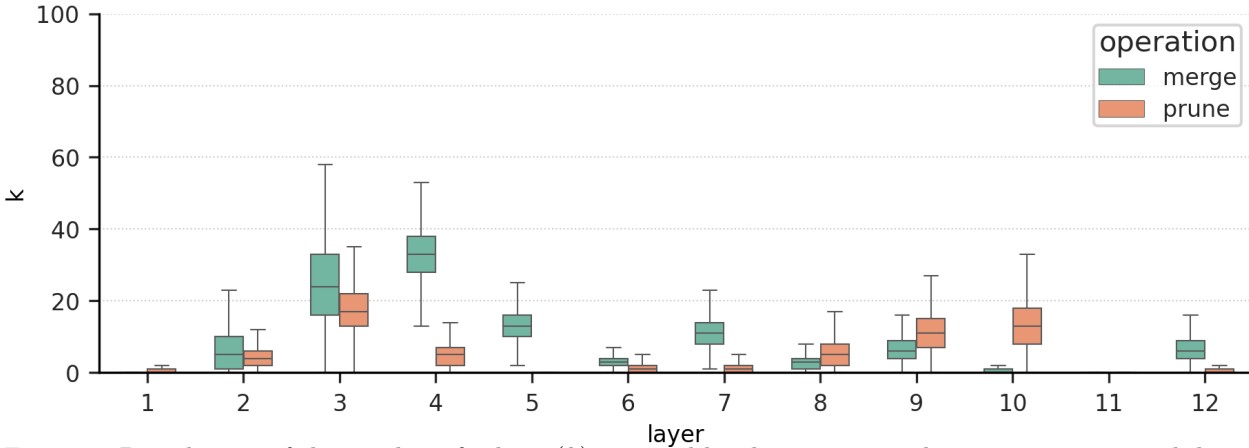

Figure 5: Distribution of the number of tokens ($k$) removed by the merging and pruning parts in each layer of LTMP DeiT-S $r_{FLOPs} \approx 0.5$.

## 4.4 Design Choices

### 4.4.1 Importance score

A key component of pruning approaches is the *importance score* used to determine which tokens to remove. The two most common choices for the importance score are:

$$\text{class attention score } s_i = \sum_{j=1}^{h} \mathbf{S}_{j0i} \tag{16}$$

where $S \in \mathbb{R}^{h \times n \times n}$ is the multi-headed extension of the attention softmax matrix (see Equation (6)) where values at index 0 correspond to the `[CLS]` token, and

$$\text{mean column attention score } s_i = \frac{1}{h \cdot n} \sum_{j=1}^{h} \sum_{k=1}^{n} \mathbf{S}_{jki} \tag{17}$$

which can be interpreted as the normalized amount that all tokens $\mathbf{x}_k$ attend to token $\mathbf{x}_i$ (Kim et al., 2022).

The results in Table 4 are obtained from DeiT-S LTP models and show that the mean column attention score performs slightly better than the class attention score, but not by a significant margin. For the remainder of this work, we use the mean column attention score (Equation (17)) as the importance score.

### 4.4.2 Merging and pruning order

As shown earlier in Table 1, the correlation between the pruning importance score and the merging similarity score is low but not zero. This means that for small reduction target values $r$, the same token might hit the thresholds for both merging and pruning. In Table 4, we compare pruning followed by merging (LTPM) with merging followed by pruning (LTMP). The results confirm that the order has no noticeable influence when the reduction rate is small, but once more tokens need to be removed LTMP is superior to LTPM. This is not surprising as merging has been shown to be more accurate than pruning (Bolya et al., 2023) and that in token merging and pruning, more tokens get merged than pruned (see Figure 5).

Table 4: Ablation of the design choices regarding the pruning *importance score* and the order in which to apply merging and pruning. All experiments are performed on DeiT-S variants.

| **FLOPs** | 2.3G | 2.7G | 3.1G |
|---|---|---|---|
| **Accuracy** | | | |
| class attention | 76.3 | 78.2 | 79.2 |
| column mean attention | 76.5 | 78.2 | 79.2 |
| LTPM | 78.4 | 79.2 | 79.5 |
| LTMP | 78.6 | 79.3 | 79.7 |

### 4.5 Ablation

Our approach has two important components: the learned thresholds and the combination of merging and pruning. In Table 5, we ablate the main components of our approach on DeiT-S for two different model sizes. We compare top-$k$ pruning and merging, where $k$ tokens are pruned in each transformer block, to learned thresholds variants. For merging, the top-$k$ approach is equal to what is used in ToMe (Bolya et al., 2023). Additionally, we compare merging and pruning individually with the combined approach.

The results show that learned thresholds improve the accuracy of pruning significantly, while for merging the improvements are only marginal. This difference in accuracy between learned thresholds and top-$k$ can be explained by examining the distribution of removed tokens as shown in Figure 6. In this box plot, we compare the distribution of the number of removed tokens with LTM and LTP with uniform top-$k$ pruning (i.e. the dashed line in the figure). As can be seen in the figure, the distribution of removed tokens with LTM is close to the uniform top-$k$ distribution, which results in many of the same tokens being merged, while for LTP the distribution is notably dissimilar.

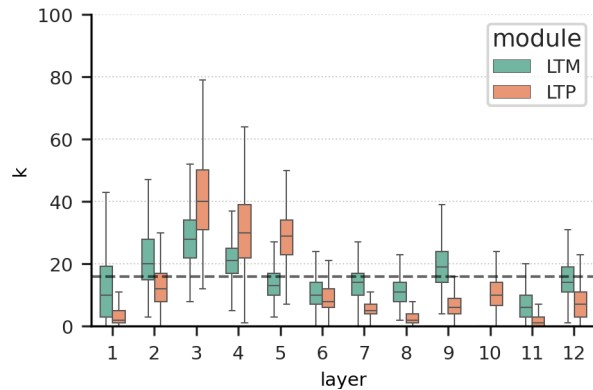

Figure 6: Distribution of the number of tokens ($k$) removed in LTP and LTM variants of DeiT-S $r_{FLOPs} \approx 0.5$.

We also observe that naively combining merging and pruning by applying both techniques with an equal fixed rate is worse than only token merging.

This ablation shows that both components are essential in LTMP. Combining merging and pruning outperforms the individual techniques but only when using the learned thresholds to balance the merging and pruning.

Table 5: Ablation of the two main components of LTMP on DeiT-S: learned thresholds and combining merging with pruning.

| Method | setting | FLOPs | Accuracy |
|---|---|---|---|
| pruning | | | |
| Top-$k$ | $k = 16$ | 2.3G | 74.8 |
| LTP | $r_{FLOPs} = 0.5$ | 2.3G | 76.5 |
| | | | |
| Top-$k$ | $k = 11$ | 3.0G | 77.9 |
| LTP | $r_{FLOPs} = 0.65$ | 3.0G | 79.2 |
| Merging | | | |
| Top-$k$ (ToMe) | $k = 16$ | 2.3G | 78.1 |
| LTM | $r_{FLOPs} = 0.5$ | 2.3G | 78.2 |
| | | | |
| Top-$k$ (ToMe) | $k = 11$ | 3.0G | 79.3 |
| LTM | $r_{FLOPs} = 0.65$ | 3.0G | 79.5 |
| Merging & pruning | | | |
| Top-$k$ | $k = 8 + 8$ | 2.3G | 77.6 |
| LTMP | $r_{FLOPs} = 0.5$ | 2.3G | 78.6 |
| | | | |
| Top-$k$ | $k = 6 + 6$ | 3.1G | 79.0 |
| LTMP | $r_{FLOPs} = 0.65$ | 3.0G | 79.6 |

### 4.6 Inference speed

Throughout this paper, we have reported FLOPs as complexity metric. While FLOPs are often regarded as a poor proxy for latency, it has also been shown that latency improvements on one type of hardware often do not translate to similar improvements on other hardware, especially in mobile and embedded devices (Bonnaerens et al., 2022). As our complexity improvements come from reducing the input tokens and both our masking modules and pruning and merging implementations are parallelized, we believe FLOPs to be the best available metric to report complexity improvements.

Nevertheless, to demonstrate our approach, we have benchmarked it on a mobile device using PyTorch's `optimize_for_mobile` function and `speed_benchmark` Android binary. Table 6 shows the latency of the baseline DeiT-S and our LTMP (with $r_{FLOPs} \approx 0.5$) reduced variant. The benchmark is conducted on a Google Pixel 7 and averaged over 200 runs (with 50 warm-up runs prior). The results show that the latency improvements, which achieve a reduction of 49.52%, are nearly identical to the theoretical FLOPs improvements, which have a reduction of 50.12%.

LTMP is also faster than ToMe while not only merging but also pruning. This likely comes from the argsort operator that is used in top-$k$ approaches such as ToMe and which is not well supported in many frameworks (Prillo & Eisenschlos, 2020). Unfortunately, despite Evit, Evo-Vit and DynamicVit having an open-source PyTorch implementation, they use operations that are not supported by TorchScript which is required for the mobile `speed_benchmark` tool.

Table 6: Latency benchmark on a Google Pixel 7.

| Method | FLOPs | Latency | Accuracy |
|---|---|---|---|
| DeiT-S (Baseline) | 4.6G | 212 ms | 79.8 |
| **LTMP (Ours)** | 2.3G | 107 ms | 78.6 |
| ToMe | 2.3G | 118 ms | 76.9 |

### 4.7 Limitations

Our learned thresholds approach requires a batch size of 1 during inference as each image is reduced differently. This is not a limitation for most resource-constrained applications as they typically operate inference with a batch size of 1. If desired, our method could be extended to accommodate larger batch sizes by either incorporating masking to a common reduction size or by converting thresholds to the average number of tokens removed per operation and layer, and applying these values in a top-$k$ adaptation. The former approach will result in higher computational complexity while the latter will result in lower accuracy.

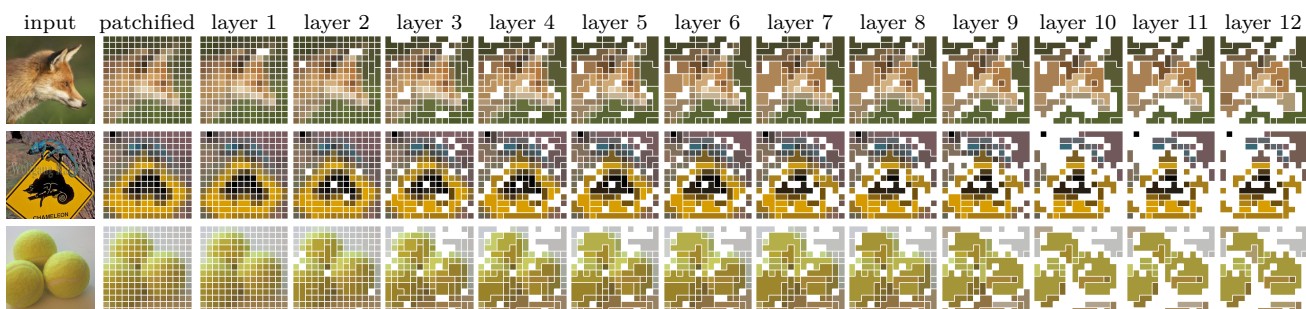

Figure 7: More visualization of the merging and pruning as applied to image patches. In every layer, the most similar tokens are merged and any unimportant tokens are pruned. The visualizations show the remaining tokens after every layer in DeiT-S.

### 4.8 Visualizations

In Figure 7, we illustrate the merging and pruning of the tokens as they are processed through the vision transformer. It can be observed how similar parts of the image get merged and how unimportant parts of the image are pruned.

## 5 Conclusion

In this work, we introduced Learned Thresholds token Merging and Pruning (LTMP) for vision transformers. LTMP makes it possible to reduce the computational cost of a vision transformer to any reduction target value with minimal loss in accuracy. LTMP adaptively reduces the number of input tokens processed by merging similar tokens and pruning the unimportant ones. Our implementation uses learned thresholds which enable different merging and pruning rates between different images and allows the model to learn the optimal trade-off between merging and pruning across layers. As LTMP only introduces two learnable parameters per transformer block, our method is able to converge within a single epoch, which is an order of magnitude quicker than other learnable approaches.

## Acknowledgments

This research received funding through the Research Foundation Flanders (FWO-Vlaanderen) under grant 1S47820N.

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

## A  ViT results

The main paper reports on DeiT models as they are most commonly used in related pruning works. However the standard pretrained ViT models as found in the `timm` library (Wightman, 2022; Steiner et al., 2022), are more accurate than the DeiT models while having the same number of FLOPs. While DeiT models are often chosen because of their more efficient training, LTMP requires only a single training epoch making the more accurate ViT models the preferred vision transformer to prune. We therefore also include results on ViT models here. Figure 8 shows the Accuracy/FLOPs trade-off curve for LTMP reduced ViT models. We also included the DeiT models as comparison. It can be observed that ViT is indeed often the better choice, except for the heavily reduced variants of the 'small' sized models, where the performance of ViT degrades faster than DeiT. More detailed listings of the results can be found in Table 7.

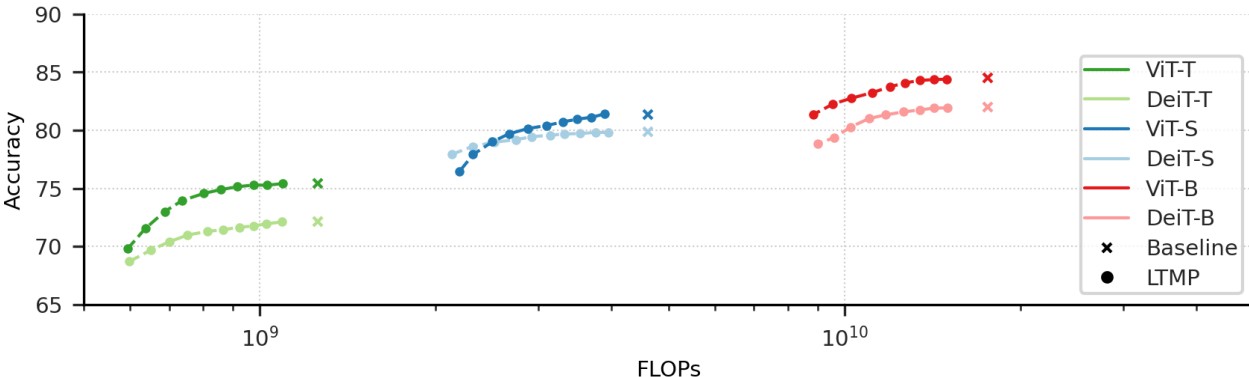

Figure 8: Accuracy/FLOPs trade-offs for LTMP variants of ViT-Tiny, -Small and -Base. DeiT variants are also plotted as comparison. FLOPs are plotted in logarithmic scale.

Table 7: Detailed results for ViT models.

| Model | $r_{\mathbf{target}}$ | FLOPs | Accuracy | Model | $r_{\mathbf{target}}$ | FLOPs | Accuracy | Model | $r_{\mathbf{target}}$ | FLOPs | Accuracy |
|---|---|---|---|---|---|---|---|---|---|---|---|
| ViT-T (Baseline) | | 1.258G | 75.45 | ViT-S (Baseline) | | 4.608G | 81.37 | ViT-B (Baseline) | | 17.583G | 84.54 |
| ViT-T | 0.85 | 1.094G | 75.4 | ViT-S | 0.85 | 3.895G | 81.4 | ViT-B | 0.85 | 15.024 | 84.5 |
| ViT-T | 0.80 | 1.029G | 75.4 | ViT-S | 0.80 | 3.689G | 81.1 | ViT-B | 0.80 | 14.258G | 84.4 |
| ViT-T | 0.75 | 0.970G | 75.3 | ViT-S | 0.75 | 3.495G | 81.1 | ViT-B | 0.75 | 13.461G | 84.3 |
| ViT-T | 0.70 | 0.911G | 75.2 | ViT-S | 0.70 | 3.309G | 80.9 | ViT-B | 0.70 | 12.685G | 84.1 |
| ViT-T | 0.65 | 0.855G | 74.9 | ViT-S | 0.65 | 3.134G | 80.5 | ViT-B | 0.65 | 11.975G | 83.8 |
| ViT-T | 0.60 | 0.803G | 74.6 | ViT-S | 0.60 | 2.915G | 80.2 | ViT-B | 0.60 | 11.152G | 83.2 |
| ViT-T | 0.55 | 0.737G | 74.0 | ViT-S | 0.55 | 2.692G | 79.8 | ViT-B | 0.55 | 10.280G | 82.8 |
| ViT-T | 0.50 | 0.688G | 73.0 | ViT-S | 0.50 | 2.501G | 79.0 | ViT-B | 0.50 | 9.559G | 82.3 |
| ViT-T | 0.45 | 0.639G | 71.7 | ViT-S | 0.45 | 2.321G | 78.0 | ViT-B | 0.45 | 8.841G | 81.3 |
| ViT-T | 0.40 | 0.600G | 70.4 | ViT-S | 0.40 | 2.248G | 76.4 | ViT-B | 0.40 | 8.465G | 80.3 |

## B  Accuracy error range

In Table 8, we report the accuracy of 5 runs with different random seeds for the main results in the paper (i.e. the data points highlighted in gray in Table 2). It can be observed that the error range on the accuracy is about $\pm 0.3$ for these data points. We, therefore, report the results in this paper with 3 significant figures.

| Model | $r_{\mathbf{target}}$ | FLOPs | Acc. #1 | Acc. #2 | Acc. #3 | Acc. #4 | Acc. #5 | Acc. range |
|-------|---------|-------|---------|---------|---------|---------|---------|------------|
| DeiT-S | 0.80 | 3.8G | 79.816 | 79.802 | 79.824 | 79.794 | 79.814 | 0.030 |
| DeiT-S | 0.62 | 3.0G | 79.598 | 79.596 | 79.584 | 79.578 | 79.564 | 0.034 |
| DeiT-S | 0.55 | 2.7G | 79.262 | 79.266 | 79.242 | 79.274 | 79.254 | 0.032 |
| DeiT-S | 0.45 | 2.3G | 78.608 | 78.604 | 78.588 | 78.574 | 78.558 | 0.050 |

Table 8: Accuracy ranges for 5 different runs of the main data points in the paper.

