# OpenReview forum: "Learned Thresholds Token Merging and Pruning for Vision Transformers"
_TMLR — Accepted by TMLR_

### Review · Reviewer_SLvV · 2023-05-20

**Summary Of Contributions:**

This paper presents Learned Thresholds token Merging and Pruning (LTMP), a new approach aimed at reducing the computational cost of vision transformers, which are effective but resource-intensive in computer vision tasks. LTMP uses dynamic modules to decide which tokens to merge or prune, enhancing performance and efficiency. Through experiments on ImageNet, LTMP was proven to achieve top-tier accuracy, quicker and with less data than prior methods.

**Audience:**

Yes

**Broader Impact Concerns:**

This work will have larger impact over the ViT community if they could address questions and issues above.

**Claims And Evidence:**

Yes

**Requested Changes:**

1. I hope the authors could better elaborate their unique contributions compared with Kim et al. 2022 and "ToMe"
2. The authors should consider studying more DeiT variants beyond DeiT-S.

**Strengths And Weaknesses:**

# Strength:
* Paper is well written.
* The authors provide comprehensive experiments.

# Weakness
* As compared by authors against Kim et al. 2022 (between Eq. 3 and Eq. 4) and “ToMe” (Section 3.3.2), the proposed method contributes marginal novelty: 1) token pruning: this work only updates tokens not removed yet (s.t. the computation is saved); however, during inference it will be the same with Kim et al. 2022; 2) token merging: Section 3.3.2 is so short because it only differs from "ToMe" by making the threshold learnable.
* The importance score is token-wise, and the similarity score is pair-wise. This means they have different numbers of values. How is their Kendall correlation  calculated in Table 1
* Variance over randomness in Table 2? The performance gap seems very close.
* Comparing with only DeiT-S is not convincing enough, as the DeiT paper proposed many other variants of ViT.

---

> ### Author Response · Authors · 2023-05-26
> **Response to Reviewer SLvV**
>
> Dear reviewer,
>
> Thank you for your review. Please find a reply to your comments below.
>
> **Weakness - (1) Novelty**
>
> While our approach is conceptually similar to Kim et al. (e.g. they also learn thresholds to prune), our approach taken to achieve this is different. Learning thresholds using the method of  Kim et al. requires multiple steps, first the thresholds are learned using a ‘soft’ pruning step and then later finetuned in a separate ‘hard’ pruning step. Additionally it requires that all original model parameters are finetuned. Our approach on the other hand allows learning the thresholds in a single finetuning scheme which can converge within a single epoch because it does not require finetuning the original model parameters.
>
> Section 3.3.2 is indeed so short as our learned thresholds approach is fundamentally the same whether pruning or merging is done.
>
> We would like to emphasize that our main contributions are the extensive combination of token merging with token pruning for which our learned threshold masking module, which allows convergence within a single epoch, is a key component (see ablation study), and our budget-aware training loss which allows us to create models of any size and allows the model to freely distribute the reduction operations across operation (merging/pruning) and layers.
> While our work extends on previously established concepts such as pruning and merging we believe that our model still has many novel contributions which resulted in top-tier accuracy for reduced vision transformers with minimal training efforts.
>
>
> **Weakness - (2) Kendall correlation**
>
> For token merging, the scoring is indeed based on pair-wise scores between tokens in sets A en B. The final similarity score of a token in A is the highest similarity score with any token in B, which allows us to rank all tokens in A.
> To compute the Kendall tau correlation we compare this similarity based ranking with the ranking of  importance scores of only the tokens in A.
>
> **Weakness - (3) Variance over randomness**
>
> We ran the specific DeiT-S model of Table 2 multiple (5) times with a different random seed and obtained the following accuracies: 79.596, 79.584, 79.578, 79.564, 79.598, all of which round to 79.6. We do agree that related work obtains similar accuracies and that LTMP does not result in a drastic increase in accuracy. Our main advantage lies in that the top-tier accuracy can be obtained with minimal training and for any desired model size.
>
>
> **Weakness - (4) Comparing only to DeiT-S**
>
> We agree with the reviewer that comparing with DeiT-S alone is not enough.
> This is why we applied LTMP to a wide range of model sizes across DeiT-T, DeiT-S and DeiT-B in Figure 4 and appendix A. Additionally the appendix also includes results on the vanilla ViT, again over a wide range of model sizes across the Tiny, Small, and Base variants.
> The reason why most of our results focus around DeiT-S is because it is the only common model on which all related work reports. This is also why Table 2 focuses on DeiT-S with a specific model size. By providing a more elaborate comparison to ToMe (the only model that does not need finetuned models to compare with) in Figure 4 and by providing accuracies on the other DeiT variants in Appendix A we hope to provide enough details for comparisons with individual works that report also on other DeiT variant.
>
> **Requested Changes - (1) clarification**
>
> Due to TMLR policy we have not yet applied any changes to the manuscript as not all reviews have been submitted. We hope that the explanation above in Weakness (1) already provides clarification of our contributions. If the reviewer wants to provide any specific clarifications they would like to see, we will make sure to include them in the updated manuscript once all reviews are submitted.
>
> **Requested Changes - (2) beyond DeiT-S**
>
> As explained above in Weakness (4), our work does study LTMP beyond DeiT-S. Could the reviewer clarify which additional variants they would like to see that are currently missing? We could also move Appendix A to the body of the paper if the reviewer prefers this.
>
> Kind regards,
>
> The authors

---

### Review · Reviewer_iwJi · 2023-06-08

**Summary Of Contributions:**

This paper presents a new method by combining token merging and pruning to reduce the computational costs of vision Transformers. Although there are several ViT acceleration methods that are based on pruning or merging, the paper explores the combination of these two groups of methods. The paper also introduced learned threshold masking modules to select the tokens to be merged or pruned effectively. The method is evaluated

**Audience:**

Yes

**Claims And Evidence:**

Yes

**Requested Changes:**

- Please refer to my comments above.

- Some minor issues. The writing of this paper can be further improved. There are some typos or grammar errors like "Our approach has two important components: the learned thresholds and the combination or merging and pruning" in Section 4.5.



**Strengths And Weaknesses:**

Strengths:
 - The paper overall is easy to follow.
 - The proposed method to select tokens is new.

Weaknesses:

- The token merging framework proposed in this paper is largely inspired by ToMe. However, although an extra token pruning method is introduced, the improvement over ToMe is not very significant. Besides, there is no direct comparison between the token merging method proposed in this paper and ToMe. Since the proposed method leverage both token merging and pruning, it is necessary to provide a comparison between token merging methods to clearly verify the effectiveness of the proposed method.

- The method cannot achieve good performance on hardware when batch size > 1.

---

> ### Author Response · Authors · 2023-06-19
> **Response to Reviewer iwJi**
>
> Dear reviewer,
>
> Thank you for your review. Please find a reply to your comments below.
>
> **Weakness - (1) comparison of LTM to ToMe**
>
> We would like to emphasize that the focus of our paper is on the combination of merging and pruning, as such our introduced components focus on enabling the model to learn how to distribute the merging and pruning operations freely across layers to achieve the desired model size.
> That being said, our ablation study does compare our learned threshold merging (LTM) with merging top-k and gives insights into why the performance of both approaches is similar. We have updated our manuscript to highlight that merging top-k is the technique used in ToMe, provided an additional data point, and extended our explanation.
>
> **Weakness - (2) batch size**
>
> The flexibility of the learned thresholds to adapt to the inputs has indeed the limitation that it requires a batch size of 1. However as explained in our limitations section, we believe that in practice this is not an issue for most resource-constrained applications which typically operate inference at a batch size of 1. If desired, it is possible to adapt LTMP to bigger batch sizes by either incorporating masking to a common (smaller) reduction size or converting thresholds to an average number of tokens removed per operation and layer and applying these values in a top-k adaption. The former approach will result in higher computational complexity while the latter will result in lower accuracy.
>
> **Requested changes - (1)  comparison of LTM to ToMe**
>
> We have updated our manuscript to highlight that merging top-k is the technique used in ToMe, provided an additional data point, and extended our explanation.
>
> **Requested Changes - (2) minor issues**
>
> We analyzed our paper with a spell and grammar checker and improved several sentences throughout the paper.
>
> Kind regards,
>
> The authors

---

### Review · Reviewer_4zvC · 2023-06-13

**Summary Of Contributions:**

This paper proposed a combination of token merge and pruning to speed up the vision transformer; furthermore, the merge and pruning thresholds are learned by simply post-training the dataset with 1 epoch. The results are on par with ToMe

**Audience:**

Yes

**Claims And Evidence:**

Yes

**Requested Changes:**

1. To better compare to ToMe, what is the performance of LTM (no P) at 3.0G? If is it still better than ToMe?
2. Following the above 1, is it possible to compare LTP with Kim et al's method?


**Strengths And Weaknesses:**

Strengths:
1. The paper is easy to follow and well-written, and the motivation is clear to explain why both merge and pruning are complement each other. Moreover, the paper proposed a unified approach to perform merge and pruning.
2. The training setup and ablation study for analysis is comprehensive.

Weaknesses:
1. The paper is the combination of two papers, which are cited in the paper: (1) ToMe for token merging and (2) LTP for token pruning.
2. When comparing to ToMe, the improvement is marginal and it requires additional 1 epoch training, which means the algo needs to have the access of the dataset rather than using it out-of-box.

---

> ### Author Response · Authors · 2023-06-19
> **Response to Reviewer 4zvC**
>
> Dear reviewer,
>
> Thank you for your review. Please find a reply to your comments below.
>
> **Weaknesses - Marginal improvement**
>
> We agree with the reviewer that while training for 1 epoch is very efficient, it still requires access to the training dataset. This is something which ToMe indeed does not require. However, we believe our accuracy improvements are not just marginal, as shown in Figure 4 in the paper, LTMP can achieve up to .5% better accuracy for DeiT-S and up to .8% better accuracy for Deit-B than ToMe for similar model sizes, which for those model sizes on ImageNet is a decent improvement. Our accuracy improvements are indeed marginal compared to other pruning works which have a fine-tuning phase, but compared to those, we improve the required fine-tuning epochs drastically.
>
> **Requested Changes - (1) LTM vs ToME**
>
> Our ablation study does compare our learned threshold merging (LTM) with merging top-k and gives insights into why the performance of both approaches is similar. We have updated our manuscript to highlight that this merging top-k is the technique used in ToMe.
> The ablation study does this for models at 2.3G.
> Additionally, we have now included data points at 3.0G and extended the surrounding text.
> While LTM does perform marginally better, further analysis confirmed our conclusion in the ablation section; the learned distribution for the thresholds when only merging is somewhat similar to uniformly k tokens in each layer.
>
> **Requested Changes - (2) LTP vs Kim et al.**
>
> We are committed to provide thorough and accurate comparisons of our work to other techniques.
> However, we believe that a comparison with Kim et al. is not feasible.
> The issue with Kim et al. is that it has no official results on ViT that we can use. We attempted to implement the approach of Kim et al. for ViT ourselves to conduct the comparison. However, even reproducing the official results on LLMs with the official code was unsuccessful as there are issues with the reproducibility of the released code (https://github.com/kssteven418/LTP/issues/9, https://github.com/kssteven418/LTP/issues/11)
> We additionally tried to implement it from scratch in our own codebase but failed to achieve accurate results. As such, any comparison we might make would be unreliable.
>
> Kind regards,
>
> The authors

---

### Author Response · Authors · 2023-06-19
**Updated manuscript**

We thank all reviewers for their effort in reading our manuscript.

We have updated our manuscript based on their suggestions in the following way:

* All results on DeiT are now in the main paper. (Appendix A is incorporated in Section 4.1)
* We improved the ablation study (Section 4.5) by adding data points for an additional model size and extended the text to highlight that this ablation compares LTM to ToMe.
* We made several small textual changes throughout the manuscript to improve clarity and to fix any spelling and grammar mistakes.

Kind regards,

The authors

---

### Decision · Action_Editors · 2023-07-18

**Recommendation:** Accept with minor revision

**Comment:**

The reviewers acknowledge the clarity of the exposition, the comprehensive evaluation, and the novelty of the proposed token selection strategy. Nevertheless, they all express concerns about the benefits of the proposed method over ToMe. Specifically, the reviewers argue that the improvements obtained with the proposed method are limited, particularly considering that it requires having access to the dataset, which ToMe does not. Although, the author's responses failed to convince the reviewers w.r.t. the benefits over ToMe, the AE considers the current set of experiments to provide sufficient evidence of the method's effectiveness.

The authors mention that they have updated their manuscript to highlight that merging top-k is the technique used in ToMe, provided an additional data point of comparison with ToMe, and extended their discussion of the comparison with ToMe. In addition to this, the AE requests the authors to:
- expand the discussion of the batch size influence, as done in their answer to Reviewer iwJi
- clarify the definition of the Kendall correlation, as done in their answer to Reviewer SLvV;
- incorporate a discussion of the variance w.r.t. different random seeds, mentioned in their answer to Reviewer SLvV. In this context, the AE would highly appreciate having similar evidence for other models.


**Audience:**

The topic studied in this paper is of interest to the TMLR audience.

**Claims And Evidence:**

The claims made in the submission are in general supported. However, the reviewers all express concerns regarding the benefits of the proposed solution over ToMe, particularly in terms of performance gain, considering that the proposed approach needs access to the dataset whereas ToMe does not.